# CD14 Expressing Precursors Give Rise to Highly Functional Conventional Dendritic Cells for Use as Dendritic Cell Vaccine

**DOI:** 10.3390/cancers13153818

**Published:** 2021-07-29

**Authors:** Maud Plantinga, Denise A. M. H. van den Beemt, Ester Dünnebach, Stefan Nierkens

**Affiliations:** 1Center for Translational Immunology, University Medical Center Utrecht, 3584 CX Utrecht, The Netherlands; m.c.plantinga-2@umcutrecht.nl (M.P.); D.A.M.vandenBeemt@umcutrecht.nl (D.A.M.H.v.d.B.); E.Dunnebach-2@umcutrecht.nl (E.D.); 2Princess Máxima Center for Pediatric Oncology, 3584 CS Utrecht, The Netherlands

**Keywords:** dendritic cells, DC subsets, cord blood, vaccine, functional differences

## Abstract

**Simple Summary:**

Dendritic cells are attractive candidates for immunotherapy to prevent disease recurrence in cancer patients. Dendritic cells are a plastic population of antigen presenting cells and a variety of protocols have been described to generate dendritic cells from either monocytes or stem cells. To induce long lasting immunity, dendritic cells need to have a fully mature phenotype and activate naïve T-cells. Here, we describe a good manufacturer protocol to generate very potent conventional DC-like cells, derived from cord blood stem cells via a CD14+CD115+ precursor stage. They express high levels of CD1c and strongly activate both naïve as well as antigen-experienced T-cells. Implementation of this protocol in the clinic could advance the efficiency of dendritic cell based vaccines.

**Abstract:**

Induction of long-lasting immunity by dendritic cells (DCs) makes them attractive candidates for anti-tumor vaccination. Although DC vaccinations are generally considered safe, clinical responses remain inconsistent in clinical trials. This initiated studies to identify subsets of DCs with superior capabilities to induce effective and memory anti-tumor responses. The use of primary DCs has been suggested to overcome the functional limitations of ex vivo monocyte-derived DCs (moDC). The ontogeny of primary DCs has recently been revised by the introduction of DC3, which phenotypically resembles conventional (c)DC2 as well as moDC. Previously, we developed a protocol to generate cDC2s from cord blood (CB)-derived stem cells via a CD115-expressing precursor. Here, we performed index sorting and single-cell RNA-sequencing to define the heterogeneity of in vitro developed DC precursors and identified CD14+CD115+ expressing cells that develop into CD1c++DCs and the remainder cells brought about CD123+DCs, as well as assessed their potency. The maturation status and T-cell activation potential were assessed using flow cytometry. CD123+DCs were specifically prone to take up antigens but only modestly activated T-cells. In contrast, CD1c++ are highly mature and specialized in both naïve as well as antigen-experienced T-cell activation. These findings show in vitro functional diversity between cord blood stem cell-derived CD123+DC and CD1c++DCs and may advance the efficiency of DC-based vaccines.

## 1. Introduction

Dendritic cells (DCs) provide long-lasting immunity via the priming of T-cell responses and stimulate adaptive memory against, e.g., pathogens and cancer cells. This capability makes them attractive candidates for cellular therapy. Sipuleucel-T is currently the only Food and Drug Administration-approved DC vaccine (Dendreon Corp., Seal Beach, CA, USA) against prostate cancer [1]. In general, with the lack of substantial clinical responses, there is a great need to improve the potency of DC-vaccination [2]. Although side by side comparisons in clinical trials are lacking, preclinical models suggest that the selection of a specific subset of antigen-presenting cell may influence the efficacy of the DC-vaccine [3,4]. DC ontogeny and functional specialization is the subject of intensive research. The current view on DC ontogeny is that granulocyte-monocyte-dendritic cell-precursors give rise to monocytes, common-dendritic cell-precursors (CDPs), and the recently identified DC3 [5,6,7,8]. CDPs further develop into plasmacytoid (p)DCs and conventional (c)DCs, while monocytes (mo) differentiate towards mo-DCs or macrophages [9]. The cDCs consist of a defined population of cDC1, expressing CLEC9a and XCR1 [10,11,12], and a heterogeneous cDC2 population, expressing CD1c and different levels of CD5 [13]. DC3s are classified as CD1c+CD163+ and express different levels of CD14 [5,6,8]. Without proper gating, DC3s are either mingled with monocytes based on CD14 or with cDC2s based on CD1c expression.

Although DCs share common features of T-cell activation and migration, they certainly have individual specialized characteristics. The cDC1 is particularly good at activating CD8 T-cells via cross-presentation [10,11,12]. The cDC2 is a more heterogeneous population activating mainly Th2-cells in response to, e.g., helminth infection, but is also able to stimulate Th17 and regulatory T-cells [14,15]. Th17 can also be induced by inflammatory DCs [16], although this population could possibly be contaminated by the newly identified DC3, reported to activate tissue-resident T-cells [6]. To achieve long lasting immunity using DCs in a vaccination strategy, it is of utmost importance to activate both CD4 as well as CD8 T-cells [17]. However, which DCs are superior in the induction of a lasting anti-tumor response is currently unknown [18]. In addition, the lack of selective membrane markers to isolate DC subsets hampers further studies into the diversity in function and therefore, many studies are performed using mouse models [19]. To be able to compare the function of human DC, CD34+ stem cell cultures have been developed successfully in vitro. In those cultures, the development of cDCs is Flt3L dependent [20,21], in contrast to DC3, which rely on GM-CSF throughout their development [6]. Mo-DCs are cultured with GM-CSF, with or without IL-4 [22]. Additionally, cDCs are often described to depend on certain niche factors provided by (modified) stromal support [23,24], e.g., producing Notch ligands [25,26]. However, these stromal cells are often of murine origin or maintained in a culture medium containing fetal calf’s serum (FCS), complicating the translation to clinical trials. Previously, we set up an in vitro method to generate DCs from cord blood (CB)-derived CD34+ stem cells and identified that cDC2 originates from CD115-expressing precursors. However, the reported heterogeneity of cDC2 (CD1c^+^CD5^lo-hi^) and the discovery of DC3 (CD1c^+^CD163^+^CD14^lo-hi^) allow a more in-depth study into the functional capacity of these subsets. In this manuscript, we set out to define the heterogeneity within the CD115+ progenitors, and report that CD1c++DCs develop from CD115+CD14+ progenitors, while CD115+CD14− progenitors generate CD123+DCs. In contrast to CD123+DCs, CD1c++DCs are highly mature and have potent T-cell activation capacity for both naïve, as well as antigen-experienced T-cells. Altogether, this put forward CD1c++DCs as a potent candidate for DC-based vaccine strategies.

## 2. Materials and Methods

### 2.1. CB Collection and CD34 Isolation

Umbilical cord blood (CB) was collected after an informed consent and was obtained according to the Declaration of Helsinki. The Ethics Committee of the University Medical Center (UMC) Utrecht approved these collection protocols (TC-bio 15-345). CB mononuclear cells were isolated from human CB by density centrifugation over the Ficoll-Paque solution (GE Healthcare, Chicago, IL, USA). CD34+cells were isolated from fresh CB using magnetic bead separation (Miltenyi Biotec, Bergisch Gladbach, Germany) resulting in a 80–95% pure CD34+ population after running two columns, as determined with flow cytometry.

### 2.2. CB-DC Culture

To generate CB-DC, a two-step protocol was applied. For the first week of expansion, 5 × 10^4^ CD34+ cells/mL were cultured and, when indicated, the second week, 1 × 10^5^ CD34+ cells/mL were cultured in X-VIVO 15 (Lonza, Basel, Switzerland) supplemented with Flt3L (50 ng/mL), SCF (50 ng/mL), IL-3 (20 ng/mL), and IL-6 (20 ng/mL). The 2-week GMP-grade expansion was performed in MACS^®^ Cell Differentiation Bag (CDB; Miltenyi Biotec) via a Luer lock system as described before [27]. After washing, the cells are differentiated at 1 × 10^5^ cells/mL in a differentiation medium, with X-VIVO 15 containing 5% human serum (HS) and supplemented with Flt3L (50 ng/mL), SCF (20 ng/mL), GM-CSF (20 ng/mL), and IL-4 (20 ng/mL) for another 7 days [10]. Recombinant cytokines were all obtained from Miltenyi Biotec. To induce the maturation of CYTOMIX, a combination of IL-1beta, IL-6, and TNF-alfa (all used at 10 ng/mL) and PGE2 (1 μg/mL) from Pfizer, was added to the DCs for 24 h in combination with the differentiation medium. For intracellular cytokine staining, brefeldin A (Sigma-Aldrich, St. Louis, MO, USA) was added during the last 16 h of culture. To obtain CB-DC subsets, CD115+CD14+ or CD115+CD14− precursors, were sorted after 7 (flasks) or 14 days (bags) of expansion. Cells were counted using trypan blue and stained with a progenitor staining mix. After sorting (>95% purity), cells were differentiated at 2 × 10^5^/mL in a differentiation medium as described above.

### 2.3. DC Phenotype

To assess the DC phenotype, cells were collected from the flasks or bags and washed twice with a FACS buffer (PBS, 2% FCS, 0.1% Sodium Azide (NaN_3_; Sigma-Aldrich)). Thereafter, the cells were incubated at 4 °C and stained with the appropriate antibody combinations. Antibodies were used for flow cytometry to differentiate DC phenotypes and assess the purity included: The aCD115 (9-4D2-1E4), aCD116 (4H1), aCD117 (104D2), aCD123 (6H6), aCD83 (HB15e), aCLEC10A (H037G3), aCD197 (G043H7), aCD1c (L161), aHLA-DR (L243), CD163 (GHI/61), and CD14 (M5E2) were purchased from Biolegend (San Diego, CA, USA); aCD34 (581) from Sony; aCD14 (61D3) and aCD274 (MIH1) from Thermo Fisher (Waltham, MA, USA); aCD45RA (HI100), aCD135 (4G8), aCD5 (UCHT2), aCD123 (7G3), aCD40 (5C3), aCD11c (B-ly6), and aCD141 (1A4) were purchased from BD Bioscience; Franklin Lakes, NJ, USA and aCD304 (AD5-17F6) from R&D, Minneapolis, MN, USA.

For intracellular (IC) staining, cells were washed with a FACS buffer after surface staining and treated with Cytofix/Cytoperm (Ebioscience, San Diego, CA, USA), according to the manufacturer’s protocol, followed by 30 min of incubation at 4 °C with KI-67 (Biolegend) to measure proliferation. For intracellular cytokine staining (ICCS), cells were washed with a FACS buffer after surface staining and treated with Cytofix/Cytoperm (BD Bioscience), according to the manufacturer’s protocol, followed by 30 min of incubation at 4 °C with IL-6 (MQ-13A5; Thermo Fisher), IL-10 (JES3-19F1), IL-8 (BH0814), and TNF-alfa (Mab11) from Biolegend and IL-12 (C8.6) from Miltenyi Biotec. The multiparameter analysis was performed on a FACS Canto II or LSR Fortessa II (BD Bioscience) flow cytometer. Dead cells were excluded by scatter gating or a fixable live/dead staining from Invitrogen. The analysis was performed using DIVA (BD Bioscience) or FlowJo software (Becton, Dickinson and Company, Franklin Lakes, NJ, USA).

### 2.4. Uptake Assay

CB-DC culture was incubated with 5 μg/mL BSA-FITC (Molecular Probes, Eugene, OR, USA) or BSA immune complex (IC) for 30 min at 4 °C to measure nonspecific binding or at 37 °C to measure the specific uptake. IC was prepared by adding BSA-FITC to BSA-IgG for 30 min of incubation at 37 °C. After incubation, cells were washed extensively with ice-cold PBS, 0.1% FCS, and 0.05% NaN_3_ and labelled at 4 °C with the appropriate antibodies. For conventional flow cytometry, the actual uptake was determined as the geometric mean of FITC+ cells within the DC population incubated at 37 °C minus the percentage of FITC+ cells incubated at 4 °C. The flow imaging analysis was performed using an ImagesStreamX Mark I platform (Amnis-Merck-millipore, Seattle, WA, USA). The INSPIRE software (Amnis-Merck-millipore, Seattle, WA, USA) was used for acquisition and IDEAS software (Amnis) for analysis. The machine was fully calibrated and passed all the tests prior to each acquisition of samples using the machines calibration and test scripts. A minimum of 10,000 events per sample was acquired. Compensation matrices were generated by running single stained particles or cells (i.e., single cell surface marker) and analyzed using the IDEAS software (Amnis-Merck-millipore, Seattle, WA, USA). Briefly, the cells were first plotted as the area versus aspect ratio of the brightfield images and a single cell gate drawn, followed by a focused gate and HLA-DR+ or HLA-DRint and CD11c+ gates based on the intensity of the respective fluorophore channels. The measurement of internalization was based on the Imagestream X internalization feature, defined as the ratio of intensity measured inside a cell to the intensity of the entire cell.

### 2.5. Mixed Leukocyte Reaction (MLR)

After isolation of CD34+ cells, used for the generation of CB-DC, the resultant CD34-fraction was enriched for T-cells using anti-CD3 magnetic microbeads (Miltenyi Biotec). For MLR, allogeneic T-cells (1 × 10^6^/mL) were labelled with cell trace violet (5 µM; Invitrogen, Carlsbad, CA, USA) and co-cultured with CB-DC (2 × 10^5^/mL) in a 96-well round-bottom plate (Corning) at a stimulator:responder ratio of 1:5. Unstimulated cell trace violet-labelled cells served as a negative control. After 4 days, cells were stained with CD3 (UCHT1), CD8 (RPA-T8), CD197 (G043H7), and CD45RO (UCHL-1) from BD Bioscience and analyzed using a FACS Canto II (BD Bioscience). The T-cell proliferation analysis was performed using a proliferation tool in the Flowjo software (Becton, Dickinson and Company) providing the division index.

### 2.6. WT1 Antigen Presentation

CD115-derived DCs were loaded overnight with WT1 peptivator (Miltenyi Biotec). The following day, DCs (50.000) were co-cultured with a previously generated HLA-A2–restricted WT1-specific T-cell clone recognizing the WT137–46 epitope [27] (50,000 T-cells) at a DC-to-T-cell ratio of 1:1 for 5 h in the presence of Golgi-stop (1/1500; BD Bioscience). The T-cells were subsequently stained for surface markers CD3 (UCHT1), CD8 (RPA-T8) from Biolegend, and LAMP-1 (H4A3; BD Biosciences). Then, after fixation and permeabilization, they were stained with the BD fix/perm (BD Biosciences) and labelled with IFN-gamma (4S.B3; BD Bioscience) and TNF-alfa (Mab11; Biolegend) antibodies. For 24 h, stimulation of the DCs and T-cells were co-cultured overnight and the supernatant was obtained the following day. The cells were stained for CD3 (UCHT1), CD8 (RPA-T8), and CD137 (4B4-1; Biolegend), followed by the flow cytometry–based analysis. The T-cells alone or T2 cells loaded with/without WT137–46 peptide (Think Peptides, Oxford, UK) were used as controls.

### 2.7. RNA-Sequence

For single cell RNA sequence, CD34+ cells were expanded and stained for CD115, CD34, and CD14 and single cells were sorted in a 384-well plate. Cells were quickly spun down and directly stored at −80 °C until further processing at Single Cell Discoveries. For bulk RNA sequencing, HLA-DRint CD1c++DCs were FACS sorted using ARIA II (BD). Total RNA was extracted using the RNeasy Mini Kit (Qiagen, Hilden, Germany) according to the manufacturer’s instructions. The concentration of RNA was quantified using a Qubit RNA HS assay and Qubit fluorometer (Thermo Fisher). Polyadenylated messenger RNA was isolated using Poly(A) beads (NEXTflex), and sequencing libraries were made using the Rapid Directional RNA-seq kit (NEXTflex). Libraries were sequenced at the Utrecht Sequencing Facility (USEQ) using the Nextseq500 platform (Illumina, San Diego, CA, USA), which produced single end reads of 75 bp. Reads were aligned to the human reference genome GRCh37 using STAR version 2.4.2a. Picard’s AddOrReplaceReadGroups (version 1.98, Broadinstitute Cambridge, MA, USA) was used to add read groups to the binary sequence alignment files, which was sorted using Sambamba (version 0.4.5 Tarasov, Vilella et al., 2015), and transcript abundances were quantified with HTSeq-count (version 0.6.1p1 Anders, Pyl et al., 2015)) using the union mode. Subsequently, reads per kilobase per million reads sequenced (RPKMs) were calculated with the edgeR RPKM function.

Differentially expressed genes were identified using the DESeq2 package with standard settings. Genes with an absolute log2 fold change larger than 0.6 and adjusted P-values of less than 0.1 were considered to be differentially expressed genes. The detailed composition of all these gene sets is provided in Appendix A. The GO terms of candidate DEGs were analyzed using ToppGene [28]. The datasets will be deposited in a publicly available database and the accession numbers will be provided during the review.

### 2.8. Statistical Analysis

Statistical analysis was conducted using the GraphPad Prism 8 software (GraphPad, La Jolla, CA, USA). The data are analyzed using the Mann-Whitney test or by the analysis of variance (ANOVA), when appropriate. When significant effects were detected, single differences between the groups were analyzed by post-hoc multiple comparison tests (Holm-Sidak or Sidak, as indicated in each figure). The level of significance was set at *p* < 0.05. The data within each group are presented as mean ± SD. Sample sizes for each experiment are indicated in the figure legends.

## 3. Results

### 3.1. Heterogenic CD14 Expression within CB-Derived CD115+ DC Precursors

We previously reported cDC2, originating from CD115-expressing precursors in a CB-stem cell DC-generation model. Here, we first set out to identify potential heterogeneous DC precursors within the CD115+ cells. We performed single cell RNA sequencing (scRNAseq) using the SORT-Seq protocol followed by sequencing of ~30 million paired end reads/plate [29]. After expansion of CB-derived CD34+ cells, we index-sorted CD115+ cells to discover precursor populations. In the subsequent validation phase, we additionally profiled CD115+ cells from three donors in total. A clustering analysis performed with the Seurat pipeline [30,31], revealed nine different groups of cells (Figure 1A). The most prominent cell clusters 0 and 1, cluster 5 and the smaller clusters 7 and 8, were enriched in transcriptional activity and cell cycle genes were observed using the gene ontology (GO)-term enrichment. They likely represent multipotent hematopoietic progenitors, but a specific identity remained ambiguous. Cluster 4 comprises genes associated with immune regulation. Cluster 2, 3, and 6 seem a continuum of discrete states of maturation rather than separate phenotypic entities, enriched in cell activation signatures (Appendix A). Interestingly, cluster 6 cells were distinguished amongst others by their expression of CD14 (Figure 1B), which is confirmed at the protein level within the CD115+ population as well as within cluster 6 using index sorting (Figure 1C and Appendix A). After expanding CB-derived CD34+ stem cells for 7 days, the majority of the CD115+ precursor lacks CD14 expression (Figure 1D). To further explore their identity and functional specialization, we isolated CD115+CD14− and CD115+CD14+ progenitors and differentiated them in the presence of Flt3L, SCF, GM-CSF, and IL-4 for an additional week.

### 3.2. CD14 Discriminates Two Distinct Differentiated CD1c Populations

To assess the proliferative capacity of each isolated precursor, the cells were stained over time with Ki67 during differentiation (d9, d12, d14) and after maturation (d15). CD115+CD14− (hereafter CD14−) cells proliferate directly and substantially, but loose potential over time. CD115+CD14+ (hereafter CD14+) cells proliferate marginally, but throughout the culture time (Figure 2A). Although variability in expansion is observed between donors, CD14− cells expanded 6 (1–10.5) times compared to 2 (1.3–4.7) times for the CD14+ cells after a total of 7 days of differentiation (Figure 2B). After differentiation, we could clearly identify different populations discriminated based on HLA-DR expression. CD14− precursors primarily generated DCs (HLA-DR+CD11c+), while CD14+ precursors differentiated into both DCs and a population expressing intermediate levels of HLA-DR (Figure 2C). The extended analysis using flow cytometry further revealed different expression patterns of other surface molecules within these CD14− derived DCs (grey) and HLA-DR^int^ DCs (black). Both cultures contain cells that express markers corresponding to cDCs, such as CLEC10A and CD1c (Figure 2D). CD14− derived DCs express CD123 (as do cDC precursors), and are therefore called CD123+DC. In contrast, CD14+ derived, HLA-DR^int^ cells do not express CD123, but express significantly high levels of CD1c (such as cDC2s), hereafter called CD1c++DC (Figure 2E). HLA-DR expression is a hallmark of DCs and therefore we set out to confirm the identity of the HLA^int^ CD1c++DCs using bulk RNA sequencing. Over 14,000 genes were found (Appendix A), of which the 2000 highest expressed were predominantly characterized in the MHC I/II-like Ag-recognition domain using the ToppGene analysis. For instance, in the top 200 expressed genes (>6000 counts); CD74, HLA-DR, CD1a, and CD1c were found. CD1c was also observed in a high proportion on the protein level (Appendix A). Although the transcriptomic analysis did not reveal the type of DC subset, it strongly suggests cDC2 or three characteristics (e.g., CLEC10a, CD1c, FceR1).

### 3.3. CD123+DC and CD1c++DC Exhibit Functional Differences

Next, the DCs from the two cultures were pulsed with either BSA alone or BSA-immune complexes (BSA-IC) coupled to FITC to measure their uptake capacity. CD123+DC showed a consistent degree of uptake of both BSA (pinocytosis) as well as BSA-IC (receptor mediated endocytosis), while CD1c++DC hardly ingested BSA or the IC (Figure 3A). The internalization intensity was greatly enhanced in the CD123+DC compared to the CD1c++DC, as confirmed using the image stream (Figure 3B).

Interestingly, the expression of HLA-DR on immature CD1c++DC increased tremendously after overnight stimulation with an inflammatory cytokine mix (cytomix), even to a similar extent as on CD123+DC. Maturation drastically increased the levels of CD83, CD40, and CCR7 in both populations (Figure 3C), albeit somewhat lower on CD1c++DC compared to CD123+DC. Moreover, PD-L1 expression is lower in CD1c++DCs suggesting a favorable co-stimulatory profile for the CD1c++DC (Figure 3D). Upon maturation, no clear differences were observed in IL-6, IL-10, IL-12, and TNFa production (Figure 3E). Altogether, CD1c++DCs have a highly matured phenotype with lower levels of PD-L1, which could potentially lead to sustained T-cell activation.

### 3.4. CD123+DCs and CD1c++DCs Have a Distinct T-Cell Stimulation Capacity

We tested the capability to activate naïve T-cells using a mixed leukocyte reaction. We cultured mature CD123+DCs and CD1c++DCs and added allogeneic T-cells from a different CB donor. After a 4-day co-culture, a division index was calculated indicating the average number of cell divisions of one cell in the original population. Although both subsets activated naïve T-cells, CD1c++ stimulated CD4 as well as CD8 T-cells substantially more (Figure 4A). Next, we questioned whether these DC subsets were able to potentiate antigen specific T-cells, using the Wilms’ tumor1 (WT1)-specific clone. DCs were pulsed with peptide to overcome the different uptake capacity by the DC subtypes. WT1-specific T-cells stimulated by peptide-pulsed CD1c++DCs produced significantly higher levels of LAMP-1 after 5 h of stimulation and increased CD137 expression after 24 h, indicating antigen-specific and cytotoxic stimulation (gating strategy Appendix A and Figure 4B).

### 3.5. Translation towards a Clinical Grade Protocol

Next, we adapted our previously developed good manufacturing practice (GMP) protocol for generation of a CB-DC vaccine to test the potential of this novel DC-product [27]. CD34+ stem cells were expanded in closed culture bags to enable sufficient expansion. After 2 weeks of culture in a bag, a similar CD115 expression including CD14 heterogeneity, was observed as described in culture flasks (Figure 5A). An expansion factor was calculated dividing the cell count obtained after a week of culture by the cell count at the start of the week. In contrast to the culture in flasks, no difference in the expansion rate was observed after sorting the CD14− and CD14+ populations after 2 weeks, followed by differentiation in bags (Figure 5B). The CD14+ generated a HLA-DR^int^ expressing population with high levels of CD1c, while the CD14− precursors developed into CD123 expressing DCs (Figure 5C). After 24 h of maturation, DCs upregulated CD83, as well as PD-L1. As similarly seen in the flask culture, CD123+DCs seem to upregulate the PD-L1 with higher intensity. In conclusion, this phenotype resembles the pre-clinical protocol to a great extent and shows the feasibility of the protocol for clinical use.

## 4. Discussion

Although the safety of many trailed DC vaccines is warranted, the clinical efficacy of DC-based vaccines should be improved. Previously, we developed a GMP-protocol to culture sufficient CB-derived DCs for vaccine purposes [27,32]. We found that these conventional DCs were originating from CD115+ precursors [33]. Here, we identified CD14-expression during the development at the progenitor stage as a discriminator between CD123+DCs and CD1c++DCs differentiation in the in vitro CBDC cultures from CD34+ stem cells. CD1c++DCs are highly mature, express lower levels of PD-L1, and excel in both naïve, as well as antigen experienced T-cell activation. In contrast, CD123+DCs are prone in the uptake of antigens, but have low abilities to cross present. This culture protocol enables a direct comparison of DC subsets and identifies functional divergence.

CD14 was identified as a differentially expressed gene (cluster 6) compared to the other clusters, although also present in clusters 2 and 3. This might be caused by a delay in differentiation of the cells found in the two other clusters, which could explain the generation of multiple populations based on HLA-DR and CD11c expression, observed after 1 week differentiation of CD14+ precursors. CD14 is used for decades to typify and isolate monocytes, but, when omitted in lineage staining, was found to be expressed on CD19-CD1c+ cells, as well. The ontogeny and function of these CD1c+ cells expressing CD14 have recently been described on a genetic level, ending up between monocytes and DC2s and were named DC3 [5,6,7,8]. In addition, the authors suggested CD88 as a better discriminator to identify monocytes [5,6,7]. The CD14+ progenitors described in this manuscript do not express CD88 (not shown). As reported previously that CD115+ cells develop into DCs [33], the other clusters most likely resemble different myeloid precursor stages, e.g., confirmed with the detection of MPO, a marker for myeloid commitment [34]. Cluster 4 differentially expresses CLEC10A, a marker previously identified on monocytes and moDC in low levels [35,36], but nowadays recognized on all human CD1c+ cDC2s [37,38]. Hence, CLEC10A can be used to classify human cDC2, with modest expression on monocytes and moDCs to consider. The CLEC10A expressed before differentiation might reveal somewhat more mature precursors that start to upregulate genes associated with DC3s and cDC2s, such as CLEC10A and CD74 [6,8]. To further elaborate on the expression of CD14 within the CD115 progenitors, a similar pattern is observed as in blood CD1c+DCs. Blood CD1c+CD14+ subsets show higher expression of CD64, CD163, and S100A8/9, characteristics of monocyte-derived cells. In addition, these CD1c+CD14+ cells express high levels of PD-L1 [39]. As a consequence, they are less able to stimulate T-cell proliferation using mixed leukocyte reaction (MLR) [39]. All these features are shared with the CD123+DCs described here. CD1c+CD14+ cells are detected in a variety of cancers, such as ovarian, breast cancer, and melanoma [16,39,40,41]. In melanoma patients, the frequency of the CD1c+CD14+ cells increased, and for the DC vaccination strategy used in these patients, CD1c+CD14+ cells were omitted from the optimized protocol and might be even a potential target to improve immunotherapy [39,42]. Bourdely et al. could align the identified DC3 with the CD14 expressing CD1c DCs. In contrast to the previously published data, they found that activated DC3 induced CD4 as well as CD8 T-cell responses and had a specific ability to induce CD103 expression on CD8 T-cells, a feature of tissue-resident memory T-cells [6]. Further studies are needed to determine if DC3 induced T-cell priming can be attributed to CD1c+CD163+CD14− in contrast to the reported CD1c+CD14+, of which both are present within DC3s. Previous data did not include phagocytosis assays to compare cDC2 to DC3. Here, we detected the lower capacity of uptake in CD1c++DCs compared to CD123+DCs. Blood cDC2 were reported to take up the antigen, but in these studies cDC2 were identified by CD1c expression and not yet separated into cDC2 and DC3 [43,44]. Similar to our findings, CD14+CD1c+ found in melanoma patients showed a higher degree of uptake compared to CD1c+ DCs [39]. It could be that the DCs are already downregulating genes involved in the uptake, while upregulating genes to obtain a mature phenotype, although no CD83 is detected on CD1c++DCs after differentiation. The receptor mediated uptake might be hampered by the lack of mannose receptor for instance (not shown). After maturation with an inflammatory cytokine mix, containing IL-1b, IL-6, TNF-alfa, and PGE2, a different pattern of maturation is seen. CD1c++DCs drastically upregulate HLA-DR and CD83, but retain inhibitory receptor expression, such as PD-L1. CD123+DCs fully mature, but simultaneously start to express high levels of PD-L1, which could hamper strong T-cell activation. Variable levels of PD-L1 are observed on DC subsets and are upregulated in IL-10 rich conditions [45]. After maturation, both CD123+DCs and CD1c++DCs produce IL-10, next to IL-12. IL-10 has been reported to influence DC maturation and T-cell activation [46,47]. Most often with a suppressive function to circumvent over activation and exhaustion [48], however upregulation does not directly correlate with suppression. CD123+DCs most likely hamper the T-cell response by their high levels of PD-L1. When CD34 is expanded in a closed culture bag for 2 weeks, the majority of the CD115 expressing cells is CD14 positive. This might be caused by differences in plastic or the CO_2_/O_2_ balance. This is beneficial, since a further selection on CD14+ precursors can be omitted, making this a fast translatable strategy for the generation of a DC-vaccine. Further understanding of Ag uptake, utilized by these CD1c++DCs, will aid in the choice of Ag formulation for loading DC subsets for vaccine development.

## 5. Conclusions

In a setting of low tumor burden, e.g., after conditioning followed by stem cell transplantation, DCs play an important role in long-term, anti-tumor T-cell immunity to overcome relapse. Here, we describe the generation of a potent CB-derived conventional DC, which strongly activates allogeneic T-cells, as well as tumor antigen-specific T-cells. The protocol not only sheds light on the ontogeny of the CB-DCs, it is optimized for clinical implementation, as well. This would greatly support the field of DC therapeutic strategies, since it is very challenging to obtain sufficient conventional DCs for vaccination.

## Figures and Tables

**Figure 1 cancers-13-03818-f001:**
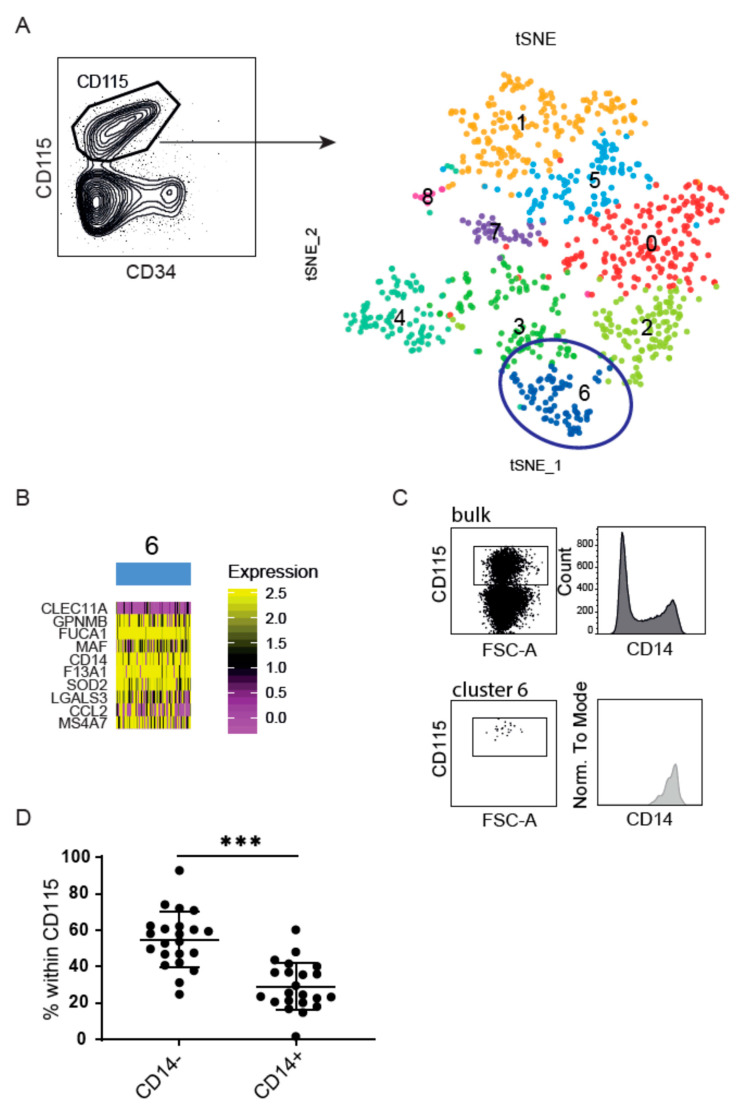
CD14 heterogeneity within the CB-culture. (**A**) The *t*-SNE analysis of CD115+ precursors (two validation samples). Colours indicate unbiased clusters. Each dot represents an individual cell. (**B**) Heat map reports for a top 10 of differentially expressed genes for cluster 6, defined in A. Colour scheme is based on *z*-score distribution from 0.0 (purple) to 2.5 (yellow). (**C**) Histogram of CD14 protein expression within bulk CD115+ population (top) or cluster 6 (bottom). For scRNAseq 1, the initial donor was used for initial analysis, and two donors (shown here) for validation. CD14 expression within CD115+ precursors after 1 week expansion of CB-derived CD34+ stem cells. (**D**) Percentage of CD14+ and CD14− populations gated within CD115+ precursors. Plots represent the mean with SD of 22 different donors. Dots represents individual data. The Mann-Whitney test was used to test significance *** = *p* < 0.001.

**Figure 2 cancers-13-03818-f002:**
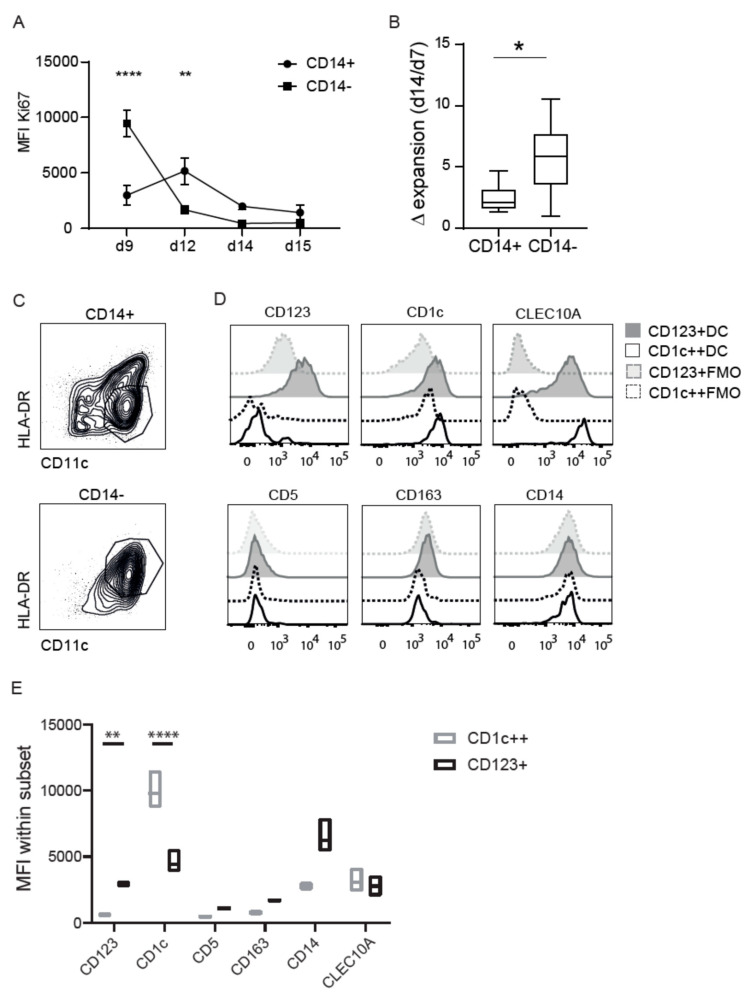
Proliferation, expansion, and differentiation. (**A**) Ki67 staining overtime in CD14− or CD14+ sorted populations in differentiation to DC in the presence of GM-CSF, IL-4, SCF, and FLt3L (d8-d14) or after maturation with PGE2, TNF-alfa, IL-6, and IL-1b in combination with the aforementioned differentiation mix (d15). The experiment was repeated 3 times; the two-way ANOVA test was used to test significance **** = *p* < 0.0001 and ** = *p* < 0.01. (**B**) Delta expansion calculated with the total differentiated cells after differentiation (day 14) divided by the sorted precursors at day 7 of expansion. Different experiments were performed with a total of eight donors. The Wilcoxon rank test was used to test significance * = *p* < 0.05. (**C**) HLA-DR and CD11c expression by differentiated CD14− or CD14+ precursors. (**D**) Flow cytometric analysis of DC markers expressed by CD1c++DCs (black), CD1c++DC FMO (black dashed), CD123+DCs (grey), and CD123+DC FMO (light grey) in histogram and (**E**) MFI in boxplot of three different donors. The populations were pre-gated on live cells based on SSC/FSC, followed by HLA-DR and CD11c shown in A. The plots represent the mean with SD. The dots represent individual data; the two-way ANOVA test was used to test significance between CD1c++DC and CD123+DC ** = *p* < 0.01, **** = *p* < 0.0001.

**Figure 3 cancers-13-03818-f003:**
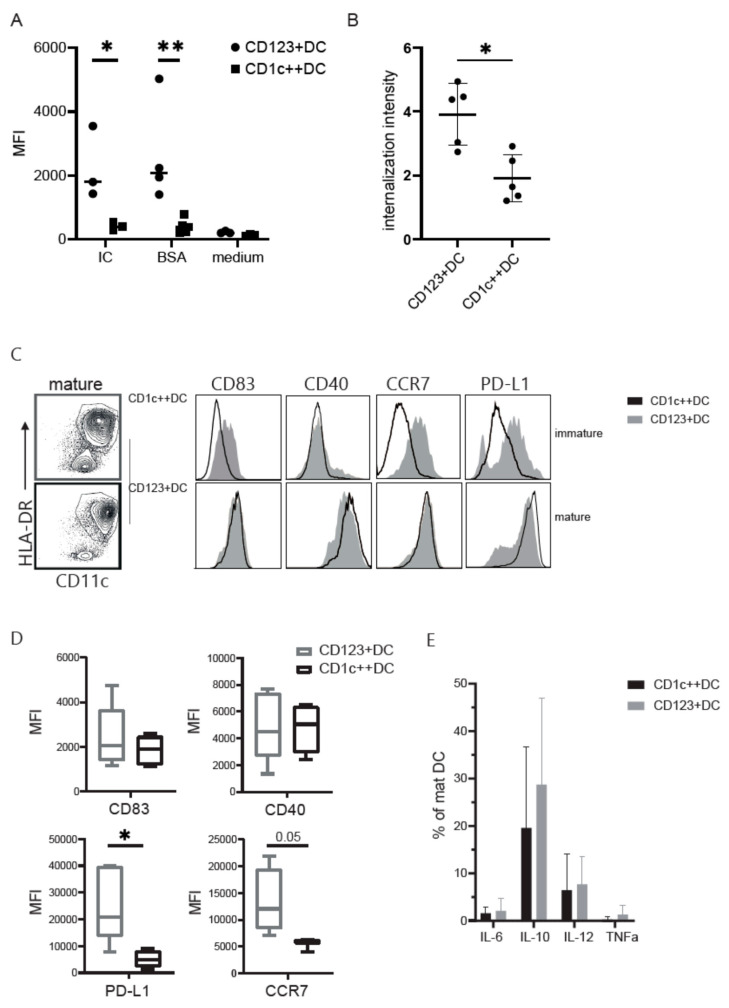
Uptake by different DC subsets and DC maturation. (**A**) Uptake of 5 ug/mL BSA-FITC or 5 ug/mL BSA immune complex (IC) by CD1c++DC and CD123+DC compared to the medium. The geometric mean of FITC^+^ signal within the DC subsets taken up at 37 °C is shown. Sidak multiple comparisons were used to test significance * = *p* < 0.05, ** = *p* < 0.01. (**B**) The internalization intensity calculated within the DC subsets after uptake of BSA-FITC. Internalization was based on the Imagestream X internalization feature, defined as the ratio of intensity measured inside a cell to the intensity of the entire cell analyzed using the IDEAS software. Data represent two independent experiments. The Mann-Whitney analysis was used to test significance * *p* < 0.05. (**C**) For the FACS analysis, live cells were gated with SSC/FSC, followed by doublet exclusion based on FSC-H/FSC-A. Gating of DC (HLA-DR+CD11c+), and thereafter CD83 and PD-L1 before and after maturation with cytomix. (**D**) Mean fluorescent intensity of CD83, CD40, PD-L1, and CCR7 on matured DC (live/single/CD11c+HLA-DR+) of the DC subsets. Significance was assessed using the Mann-Whitney test (* = < 0.05). (**E**) Intracellular cytokines (IL-6, IL-10, IL-12 (p40/p70), and TNF-alfa stained after fixation and permeabilization within DC subsets after maturation. (**D**,**E**) Experiments were performed with four different donors.

**Figure 4 cancers-13-03818-f004:**
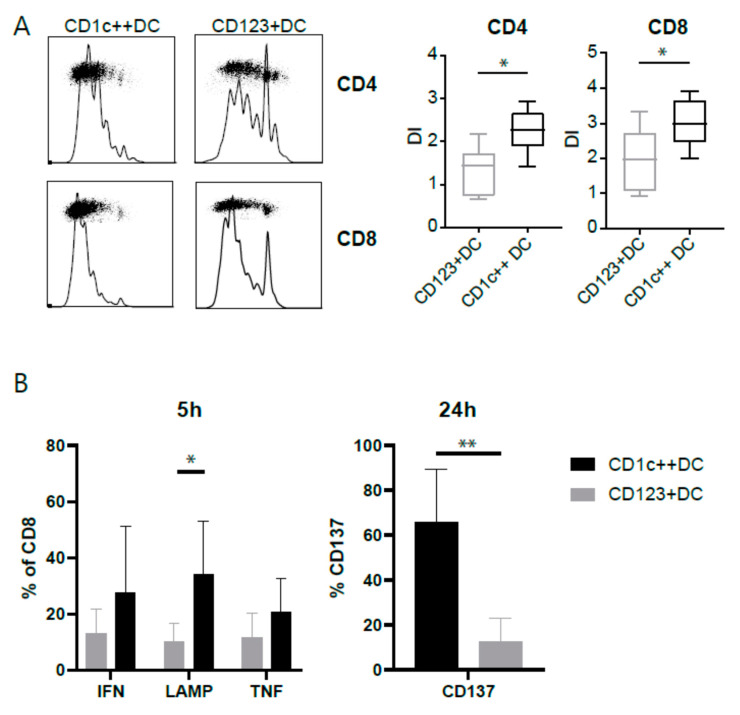
T-cell activation assays. (**A**) T-cell activation was measured in a mixed leukocyte reaction (MLR). Previously isolated CD3 T-cells from a different CB donor were thawed and labelled with a cell tracer violet dye. Cells were seeded at 1 × 10^5^ cell/well and stimulated with 2 × 10^4^ cells/well DC subsets for 4 days. Proliferation was measured by FACS and the division index (DI) was calculated using Flowjo. DI was the total number of divisions divided by the number of cells that went into the division gated within the CD4 (top and quantified top left) or CD8 (bottom and quantified top right). Six donors were used for these experiments. (**B**) Antigen specific T-cell activation by DC subsets pulsed o/n with 6 nmol of Wilm’s tumor (WT1) peptivator (Miltenyi). T-cell activation was measured by their intracellular IFN-gamma and TNF-alfa and extracellular LAMP-1 expression after 5 h or CD137 expression after 24 h of co-culture. Data represent five independent experiments. When indicated, significance was assessed using the Mann-Whitney test (* ≤ 0.05, ** ≤ 0.01).

**Figure 5 cancers-13-03818-f005:**
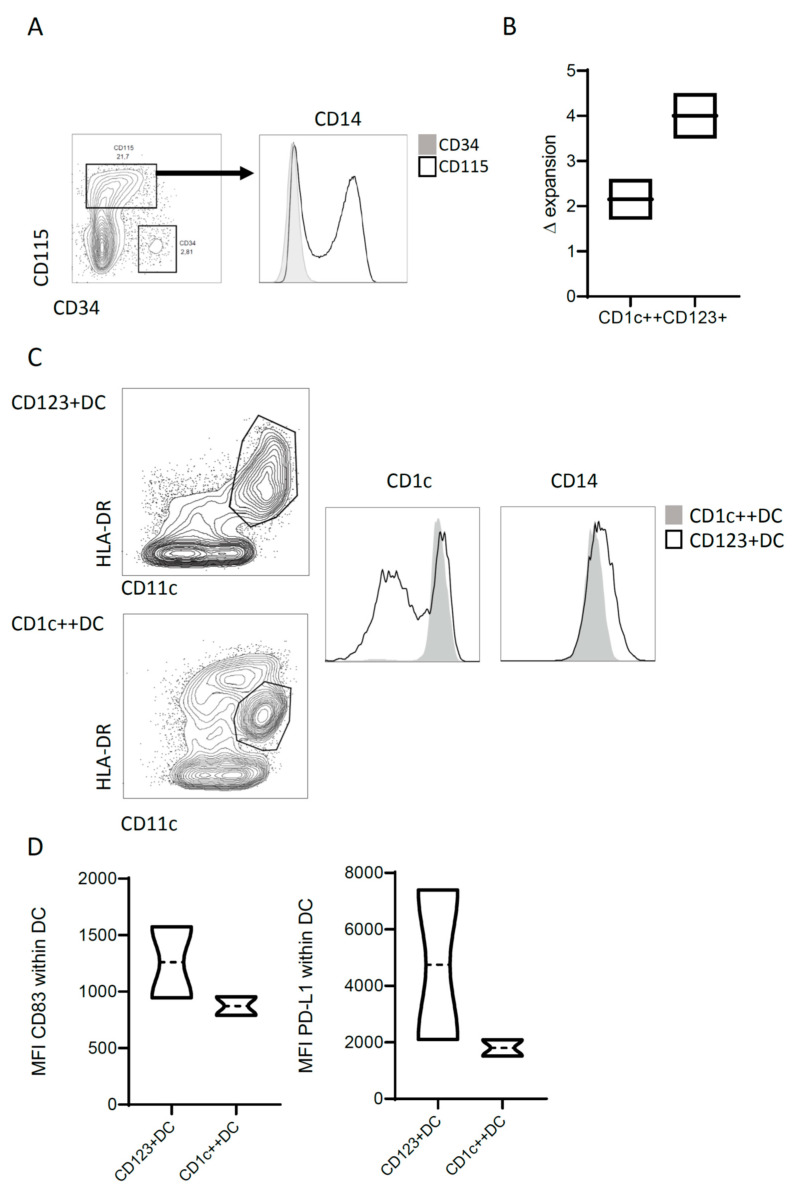
GMP translation of pre-clinical subsets. Two-week expansion of CD34+ followed by one week of differentiation to DCs. After expansion, either CD115+CD14− or CD115+CD14+ were isolated and differentiated in closed culture bags. (**A**) Phenotype of precursors at day 14. (**B**) Expansion factor calculated per week by dividing the total cells after 1 week by the starting amount of that week. (**C**) Phenotype after differentiation using flow cytometry. (**D**) Phenotype after maturation using flow cytometry. These experiments were performed with two donors.

## Data Availability

The datasets will be deposited in a publicly available database and the accession numbers will be provided during review.

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
