# Peer review of "CD14 Expressing Precursors Give Rise to Highly Functional Conventional Dendritic Cells for Use as Dendritic Cell Vaccine"

_cancers, 2021, doi:10.3390/cancers13153818_

Round 1

Reviewer 1 Report

In this article “CD14 expressing precursors give rise to highly functional conventional dendritic cells (cDCs) for use as dendritic cell vaccine”, the authors define CD115+ progenitors and show that a specific DC population, CD115+CD14-CD123+ DCs, can present significantly more antigen than CD115+CD14+CD1C++ DCs. However, CD115+CD14+CD1C++ DCs are highly mature and have a potent T-cell activation capacity for DC-based cancer vaccine, compared to CD115+CD14-CD123+ DCs. The authors also contribute to the discussion that different DC subsets can influence cancer therapy using in DC-based vaccine context.

 Some suggestions are made as follows to improve the scholarly presentation of the article. 

(1) Using CD115+CD14-CD123+ DCs and CD115+CD14+CD1C++ DCs to therapy cancer is the key concept for this article. These cells are most likely cDC2 (in human cDC2B marker: CD5-, CD14+, CD163+) populations. Without mentioning why we use cDC2, but not cDC1, to develop DC-based vaccine for cancer therapy in the first paragraph, it will be difficult to understand the context this paper in the general landscape of DC-based vaccination, as it is generally acknowledged that cDC1s are superior in antigen cross-presentation or cross-priming ability. The advantages and benefits of using CD123+ or CD1C++ cells should be described. A helpful reference can be found in the following references. [ Therapeutic cancer vaccine, Nature Reviews Cancer, pages 360–378, 2021 and Recent Progress in Dendritic Cell-Based Cancer Immunotherapy, Cancers, 20;13(10):2495).]

(2) Authors attempt to define new populations, CD115+CD14-CD123+ DCs (like cDC2) and CD115+CD14+CD1C++ DCs (like cDC2B or cDC3). Authors should give more background information about these two populations, compared to cDC1, cDC2 or cDC3.

Minor point:

(1) Please provide high-resolution images to make it easier for the editors and referees to review your manuscript.

Author Response

(1) Using CD115+CD14-CD123+ DCs and CD115+CD14+CD1C++ DCs to therapy cancer is the key concept for this article. These cells are most likely cDC2 (in human cDC2B marker: CD5-, CD14+, CD163+) populations. Without mentioning why we use cDC2, but not cDC1, to develop DC-based vaccine for cancer therapy in the first paragraph, it will be difficult to understand the context this paper in the general landscape of DC-based vaccination, as it is generally acknowledged that cDC1s are superior in antigen cross-presentation or cross-priming ability. The advantages and benefits of using CD123+ or CD1C++ cells should be described. A helpful reference can be found in the following references. [ Therapeutic cancer vaccine, Nature Reviews Cancer, pages 360–378, 2021 and Recent Progress in Dendritic Cell-Based Cancer Immunotherapy, Cancers, 20;13(10):2495).]

Thank you for this valuable comment. In the complexity of DC subsets I agree that we should label them by the described names according to recent literature. However, we here generate potent DCs to use for vaccination strategy, and do not compare the DCs to their blood counterparts (beyond the scope of this manuscript). So I am in doubt to strongly conclude that they are cDC2B, while DC3 is recently added to the family as well. I added “To achieve long lasting immunity using DCs in a vaccination strategy, it is of utmost importance to activate both CD4 as well as CD8 T-cells[17]. However, which DCs are superior in the induction of a lasting anti-tumor response is currently unknown[18]. “ to the introduction (line63-66) to touch upon the hypothesis that solely cDC1 (CD8 activators) might not be the final solution to increase DC vaccination efficacy. There is significant expression of CD1c on the one subset and CD123 on the other described in this manuscript, so I would suggest not to change the naming, unless strongly advised otherwise. 

  1. Smith, C.M.; Wilson, N.S.; Waithman, J.; Villadangos, J.A.; Carbone, F.R.; Heath, W.R.; Belz, G.T. Cognate CD4(+) T cell licensing of dendritic cells in CD8(+) T cell immunity. Nature immunology 2004, 5, 1143-1148, doi:10.1038/ni1129.
  2. Saxena, M.; van der Burg, S.H.; Melief, C.J.M.; Bhardwaj, N. Therapeutic cancer vaccines. Nat Rev Cancer 2021, 21, 360-378, doi:10.1038/s41568-021-00346-0.

(2) Authors attempt to define new populations, CD115+CD14-CD123+ DCs (like cDC2) and CD115+CD14+CD1C++ DCs (like cDC2B or cDC3). Authors should give more background information about these two populations, compared to cDC1, cDC2 or cDC3.

I agree that CD115+ derived DCs are phenotypically and genotypically similar to cDC2 (as described by us previously in Plantinga, M. et al. Cord-Blood-Stem-Cell-Derived Conventional Dendritic Cells Specifically Originate from CD115-Expressing Precursors. Cancers (Basel) 2019,11,doi:10.3390/cancers11020181). A more fundamental study could describe a more sophisticated analysis comparing CB-derived DCs described in this manuscript to blood cDC1, 2 and 3. However, in our opinion that is beyond the scope of this manuscript, which is to develop sufficient DCs to use for vaccine-strategies. Thus far, no GMP-protocol is available for sufficient cDC1 (and also CD4 stimulation), nor the newly identified and not yet fully characterized DC3. Here we identified potent DCs which will be implemented in a Phase I/II clinical trial soon, but in the new light of heterogeneity zoomed in on the CD115 population (this manuscript). Within this heterogenic cDC2-like CB-DCs we phenotypically compared a panel of markers expressed by the different blood DC subsets, and thereafter emphasis their functional differences.

In part 3.2 I added the DC subsets after the well described marker to compare them a bit further to cDC subsets. E.g. “In contrast, CD14+ derived, HLA-DRint cells do not express CD123, but express significantly high levels of CD1c (like cDC2s)” (line 267).

Minor point:

  • Please provide high-resolution images to makeit easier for the editors and referees to review your manuscript.

High-resolution images will be provided.

Reviewer 2 Report

In this manuscript the authors Plantinga et al. characterize CD14 derived dendritic cells as potential dendritic cell vaccine. This study is well executed with validated and clear scientific protocols.  Overall, this study is very descriptive and not entirely convincing.

General remark:

The simple summary is missing!

Abstract:

Although the title contains CD14-expressing precursors, these cells are not mentioned in the abstract. The terms used in the abstract for the DC subtypes are introduced only during the course of the text. The authors should therefore adapt the abstract.

Section 3.1:

In general, this part is well written and understandably. But wouldn't it be better to move the part of Figure 2 A and B into Section 3.2?

The quality of the figure is very poor. It is hard to even read the axis labeling.

The figure legend doesn’t fit tot he figure itself. For example D) is missing. Please define the p-value for ***.

Section 3.2:

Please provide a gating strategy for the characterization of CD14- and CD14+ DCs. Otherwise it is a little difficult to understand Figure 2D. Also, the authors should represent the data using i.e. offset overlays to to better demonstrate the differences between the curves.

Since CD1c seems tob e highly expressed by both subsets, is this really a suitable marker to classify the cells?

CLEC10A is missing in Figure 2E.

How many samples were analysed?

Please define the p-value for * and ****.

Section 3.3:

These experiments show clearly that there are certain differences between CD123+ and CD1c++ DCs. However, I wonder why the authors did not include unstimulated cells in the analysis. „Baseline values“ would make it possible to better classify the maturation of the cells. I don't understand how the authors can claim that expression of HLA-DR and others increases dramatically after maturation if immature cells were not tested for these markers? Therefore, the authors should compare the results with untreated cells respectively.  

I also wonder why the authors did not look at the differences in expression of costimulatory molecules, such as CD80 or CD86, which are important for effective T cell activation. Are CD80 and CD86 expressed differently by the two subtypes?

The quality of the figure is very poor. It is hard to even read the axis labeling.

How many samples were analysed in D) and E)?

Please define the p-value for * and **.

Section 3.4:

The arrangement of the groups in the graph should be adapted to Figure 3. That means first CD123 + DC then CD1c++ DC.

Representative FACS plots for % of IFN (IFNγ?), LAMP and TNF(α?) of CD8 T cells’ should be provided.

How many samples were analysed?

Author Response

General remark: The simple summary is missing!

Answer: Thank you for pointing this out. We added a simple summary as written below.

Dendritic cells are attractive candidates for immunotherapy to prevent disease recurrence in cancer patients. Dendritic cells are a plastic population of antigen presenting cells and a variety of protocols have been described to generate dendritic cells from either monocytes or stem cells. To induce long lasting immunity, dendritic cells need to have a fully mature phenotype and activate naïve T-cells. We here describe a good manufacturer protocol to generate very potent conventional DC-like cells, derived from cord blood stem cells via a CD14+CD115+ precursor stage. They express high levels of CD1c and strongly activate both naïve as well as antigen-experienced T-cells. Implementation of this protocol in the clinic could advance efficiency of dendritic cell based vaccines.

Abstract:

Although the title contains CD14-expressing precursors, these cells are not mentioned in the abstract. The terms used in the abstract for the DC subtypes are introduced only during the course of the text. The authors should therefore adapt the abstract.

Answer: We have changed the abstract and added the CD14-expressing precursors in line 26-29 and pointed out here.

We here performed index sorting and single-cell RNA-sequencing to define heterogeneity of in vitro developed DC precursors and identified CD14+CD115+ expressing cells developing into CD1c++ DCs and the remainder cells brought about CD123+ DCs, and assessed their potency.  

Section 3.1:

In general, this part is well written and understandably. But wouldn't it be better to move the part of Figure 2 A and B into Section 3.2?

The quality of the figure is very poor. It is hard to even read the axis labelling.

The figure legend doesn’t fit to the figure itself. For example D) is missing. Please define the p-value for ***.

Answer:

For an even higher readability we moved the part of figure 2A and 2B into Section 3.2. Indeed the quality of the figures can be improved, which will be taken care of uploading them separately as pdf instead of pasted within the word file. The size of the axes will be enlarged. This will be the case for all figures.

Figure legends has been adapted. C is added and C has been changed to D. We added the p-value (***= p<0.001)

Section 3.2:

Please provide a gating strategy for the characterization of CD14- and CD14+ DCs. Otherwise it is a little difficult to understand Figure 2D. Also, the authors should represent the data using i.e. offset overlays to to better demonstrate the differences between the curves.

Since CD1c seems to be highly expressed by both subsets, is this really a suitable marker to classify the cells?

CLEC10A is missing in Figure 2E.

How many samples were analysed?

Please define the p-value for * and ****.

Answer: Thank you for your comments adding to the readability of our manuscript. The gating strategy is pointed out in figure 2C. This has been better clarified in the text (line 261). Also figure 2D is changed to improve the visibility of the differences between the curves. Per accident an older figure has been pasted in the figure, this has been changed including CLEC10A and updated statistics. For this experiment at least 3donors are used and 8 donors for expansion in figure 2B. P-values has been defined in the figure legends.

Furthermore, we agree that both DC subtypes express CD1c, although DCs generated from the CD14+ precursors express CD1c significantly high, while the DCs generated from the CD14- precursors express higher levels of CD123, which we therefor used to identify the DC subtypes. It would perhaps be more sophisticated to compare levels to the blood DCs (CD123+ precursors/pDCs, CD1c cDC2 etc), but this is beyond the scope of this manuscript to generate cDC for the use of vaccination. 

Section 3.3:

These experiments show clearly that there are certain differences between CD123+ and CD1c++ DCs. However, I wonder why the authors did not include unstimulated cells in the analysis. „Baseline values“ would make it possible to better classify the maturation of the cells. I don't understand how the authors can claim that expression of HLA-DR and others increases dramatically after maturation if immature cells were not tested for these markers? Therefore, the authors should compare the results with untreated cells respectively.  

I also wonder why the authors did not look at the differences in expression of costimulatory molecules, such as CD80 or CD86, which are important for effective T cell activation. Are CD80 and CD86 expressed differently by the two subtypes?

The quality of the figure is very poor. It is hard to even read the axis labeling.

How many samples were analysed in D) and E)?

Please define the p-value for * and **.

Answer: We agree that unstimulated cells provide a better understanding of maturation status of the DCs, and included unstimulated cells in figure 3C. Thereafter we compare the DC subtypes to each other. Unfortunately we did not stain the mature cells for CD80 and CD86 in these experiments although we previously showed comparable levels of CD80 and CD86 to CD83 in the unsorted bulk culture (ref 30 de Haar, C., et al., Generation of a cord blood-derived Wilms Tumor 1 dendritic cell vaccine for AML patients treated with allogeneic cord blood transplantation. Oncoimmunology, 2015. 4(11): p. e1023973.) 

Figure quality will be improved. Amount of samples (4) is added to the figure legends as well as the significance *=p<0.05 **=P<0.01.

Section 3.4:

The arrangement of the groups in the graph should be adapted to Figure 3. That means first CD123 + DC then CD1c++ DC.

Representative FACS plots for % of IFN (IFNγ?), LAMP and TNF(α?) of CD8 T cells’ should be provided.

How many samples were analysed?

Answer: We rearranged the groups in figure 4 to first CD123+DCs followed by CD1c++ DCs. Representative FACS plots and gating strategy has been added to supplementary figure 3. For the MLR 6 different donors were used, for the WT1 assay at least 5 donors (5donors for 5h assay, 6 for 24h assay).

Round 2

Reviewer 2 Report

Thank you for addressing my comments. I think that the quality of the manuscript has increased. Nevertheless, the authors should add information about the number of samples / donors examined to each figure before publication. Please also indicate whether Figure 3D represent results of immature or mature cells. That is not entirely clear from the text. Many Thanks.

Author Response

Thank you for addressing my comments. I think that the quality of the manuscript has increased. Nevertheless, the authors should add information about the number of samples / donors examined to each figure before publication. Please also indicate whether Figure 3D represent results of immature or mature cells. That is not entirely clear from the text. Many Thanks.

Answer: I added the number of samples/donors to each figure. In figure 3D I added mature: Mean fluorescent intensity of CD83, CD40, PD-L1 and CCR7 on matured DC (live/single/CD11c+HLA-DR+) of the DC subsets. Thank you for pointing this out.